# Clinical Outcomes of the Intraocular Lens Injector and Busin Glide for Descemet Stripping Automated Endothelial Keratoplasty in Patients with Iridocorneal Endothelial Syndrome

**DOI:** 10.3390/jcm12051856

**Published:** 2023-02-26

**Authors:** Saiqing Li, Zihao Liu, Binjia Sun, Zelin Zhao, Haiou Wang, Qinxiang Zheng, Wei Chen

**Affiliations:** 1National Clinical Research Center for Ocular Diseases, Eye Hospital, Wenzhou Medical University, Wenzhou 325027, China; 2The Eye Hospital and School of Ophthalmology and Optometry, Wenzhou Medical University, Wenzhou 325027, China

**Keywords:** Descemet stripping automated endothelial keratoplasty, iridocorneal endothelial syndrome, graft inserter, endothelial cell damage, safety

## Abstract

**Purpose:** To report the outcomes of Descemet stripping automated endothelial keratoplasty (DSAEK) performed in iridocorneal endothelial (ICE) syndrome patients using the intraocular lens injector (injector), in comparison with those using the Busin glide. **Methods:** In this retrospective, interventional comparative study, we evaluated the outcomes of DSAEK performed using the injector (*n* = 12) or the Busin glide (*n* = 12) for patients with ICE syndrome. Their graft position and postoperative complications were recorded. Their best-corrected visual acuity (BCVA) and endothelial cell loss (ECL) were monitored over a 12-month follow-up period. **Results:** DSAEK was conducted successfully in the 24 cases. The BCVA improved from the preoperative 0.99 ± 0.61 to 0.36 ± 0.35 at 12 months after operation (*p* < 0.001), with no significant difference between the two groups (the injector group and the Busin group) (*p* = 0.933). ECL at 1 month after DSAEK was 21.80 ± 15.01% in the injector group, which was significantly lower than 33.69 ± 9.75% of the Busin group (*p* = 0.031). No surgery-related complications were observed in the 24 cases intraoperatively or postoperatively except that one case suffered from postoperative graft dislocation, without statistical difference between the two groups. **Conclusions:** At 1 month after surgery, the use of graft injector for delivering DSAEK-based endothelial graft may cause significantly less endothelial cell damage than the pull-through technique used in the application of Busin glide. The injector allows safe endothelial graft delivery without the need of anterior chamber irrigation, which increases the ratio of successful graft attachment.

## 1. Introduction

Iridocorneal endothelial (ICE) syndrome, a rare and challenging eye syndrome, includes progressive essential iris atrophy, the Cogan–Reese syndrome, Chandler’s syndrome and mixed forms. Pathologically, the abnormal corneal endothelial cells migrate to the anterior chamber angle, the iris, and possibly the zonula fibers, which causes corneal edema, corneal endothelial failure, broad peripheral anterior synechiae, iris stromal alterations and secondary glaucoma [1,2,3,4,5], among which corneal edema and glaucoma are the main therapeutic problems.

Keratoplasty is indicated for the treatment of irreversible corneal edema and corneal endothelial failure in ICE syndrome [6,7,8]. Compared to penetrating keratoplasty (PK), endothelial keratoplasty (EK) is more preferable for ICE syndrome since the endothelium is the layer that needs to be replaced. Descemet stripping automated endothelial keratoplasty (DSAEK) is a less-invasive method featuring faster recovery, better visual outcome and higher endothelial cell survival [9,10,11,12,13]. However, EK may be technically more challenging for surgeons when treating ICE syndrome because of the complicated condition in the anterior segment [6,14]. In addition, the minimal loss of the endothelial cells is the most crucial issue that should be secured during surgery, which depends on the size of the incision, and the method of graft delivery [15]. Tsatsos et al. compared the injector device and grasping forceps for graft implantation in Descemet stripping endothelial keratoplasty, and reported that the injector technique will lead to higher endothelial cell survival at 6 and 12 months after surgery [16]. Moreover, the use of injector yielded a low ratio of intraoperative complications for DSAEK [17]. Based on these findings in cases with Fuchs endothelial dystrophy and bullous keratopathy [17,18,19], we speculate that the injector technique may exert excellent outcomes in treating intractable diseases such as ICE syndrome. Thus, in the current study, we improved the injector technique via a small incision and stable anterior chamber (AC) for DSAEK, and enrolled 24 ICE syndrome cases to compare the outcomes of the injector and Busin glide in delivering the endothelial graft.

## 2. Materials and Methods

This retrospective, interventional comparative study included a total of 24 consecutive patients with endothelial dysfunction secondary to ICE syndrome who underwent DSAEK at the Affiliated Eye Hospital of Wenzhou Medical University between January 2018 and December 2021. All the patients finished a one-year follow-up period. The study adhered to the tenets of the Declaration of Helsinki, and was approved by the Institutional Review Board of the Affiliated Eye Hospital of Wenzhou Medical University under No. 2020-219-K-199-01. All the participants provided a written informed consent after receiving an explanation of the risks/benefits of the study.

The ICE syndrome was confirmed by the presence of at least 2 of the following 3 principal diagnostic criteria [20]: unilateral corneal edema caused by an abnormality of the corneal endothelium or a typical unilateral hammered-silver appearance of the posterior cornea; iris atrophy with corectopia, ectropion uvea, holes, or nodules in the same eye as the corneal changes; and broad peripheral anterior synechiae or other iridocorneal adhesions. All the patients presented significant endothelial dysfunction secondary to ICE syndrome before surgery. They underwent a complete evaluation including measurement of best-corrected visual acuity (BCVA), slit-lamp examination, intraocular pressure (IOP), and dilated funduscopy. Preoperatively, the donor endothelial cell density (ECD) was provided by the Eye Bank of Wenzhou using specular microscopy (CellChek D; Konan Medical, CA, USA). According to the method of graft insertion in DSAEK (intraocular lens injector or Busin glide), we divided the patients enrolled into the injector group and the Busin group.

*Surgical technique:* If only DSAEK was performed, 1.0% pilocarpine nitrate was used to reduce the pupil diameter preoperatively. When combining the DSAEK and the cataract surgery, the preoperative pupil dilation was achieved with 1.0% tropicamide, 1.0% cyclopentolate, and 2.5% phenylephrine. All surgeries were performed by the same surgeon (Wei Chen) under general anesthesia.

*Donor preparations*: The graft was prepared using automatic equipment with an artificial AC on the sterile worktable, for ensuring an appropriate depth, center positioning and uniform cutting of the graft. The central corneal thickness of the donor was measured using an ultrasonic pachymeter. A corneal trephine with a diameter of 8.5 or 8.75 mm was used to punch the donor from the endothelial side down.

*Donor insertion with the injector (n = 12)*: At the beginning of the surgery, the corneal epithelium was removed by an iris restorer. The same corneal trephine was used to make a circular mark on the surface of the recipient’s cornea with gentian violet ink. A clear corneal incision was made near the limbus at 9 o’clock using a slit-angled 3.0 mm disposable keratome (Mani Ophthalmic, Tochigi, Japan), a paracentesis was made with a 15-degree tipped blade at 4 o’clock. Then the viscoelastic was injected to maintain the AC. The trephination diameter ranges from 7.5–9.0 mm depending on the size of the corneal endothelium lesion. Corneal endothelium and Descemet membrane were dissected by reverse stripping hook.

The intraocular lens injector (Viscoject eco, Medicel AG, Wolfhalden, Switzerland) consists of an injection tube and a loading cartridge for loading the endothelial graft (Figure 1). As for the graft insertion, the folded graft was placed gently with the endothelial side up in the groove of the cartridge, which was immersed in balanced salt solution (BSS). Then, a smooth forceps was used to pull the graft into the distal aperture of the tube without inducing flexion in the scroll, close the wings of the cartridge and load it into the injection system. Then the folded graft was neatly introduced into the recipient AC through the 3 mm incision without the help of an AC maintainer. The injector inclined downward into the AC with graft the endothelial side up. Then the injector rotated 90 degrees clockwise and slowly pushed the folded graft into the AC. (Figure 2). A 10-0 suture was placed at the 3 mm incision to prevent the graft from slipping. Most grafts would unfold automatically, and if not, the viscoelastic would be injected into the AC centrally under the graft to help unfold the graft. Then the viscoelastic was removed completely and the graft was placed at the right position. In the end, an air tamponade was conducted to keep graft adherence. The surgical procedures were presented in Appendix A.

*Donor insertion with the Busin Glide (n = 12)*: The same corneal trephine was used to make a circular mark on the central epithelial surface of the recipient cornea with gentian violet ink. A temporal clear corneal tunnel was made using a 3.2 mm keratome. An AC maintainer connected to a BSS infusion line was inserted via a paracentesis at the inferonasal limbus. A long limbal tunnel was made nasally 180 degrees opposite the corneal tunnel with a microvitreoretinal blade and a further paracentesis was made at the limbus at the 8-o’clock position for the right eye and at the 2-o’clock position for the left eye using a 15-degree blade. The Descemet membrane was stained with 0.01% trypan blue dye and scored using a blunt-tip reverse Sinskey hook using the epithelial mark as a guide. The Descemet membrane was stripped off using Melles Descemet strippers and was removed with a Macpherson forceps. The incision then was enlarged to 4.2 mm for applying the Busin technique.

The donor lenticule was separated from the anterior lamellar surface and transferred onto the glide endothelial side upward, with the aid of Macpherson forceps or BSS in a Rycroft cannula, and a small amount of viscoelastic was placed on the endothelial surface. The Busin forceps (Moria SA, Antony, France) was inserted into the glide upward to the slot to grasp the donor button, which then was pulled into the glide opening. The glide then was inverted and positioned at the entrance of the corneal tunnel. The Busin forceps were passed via the nasal incision to grasp the edge of donor lenticule and to pull it into the AC (Figure 3). After graft delivery into the anterior chamber, an air tamponade was conducted to ensure good graft–host apposition.

All the patients were required to maintain an upward-facing position on their beds for at least 6–8 h immediately after operation. Postoperatively, all the patients received antibiotic steroid eye drops 4 times every day for the initial 3 months. The eye drops were tapered gradually over a period of 6 months. If the patients had a high intraocular pressure, medication was given to reduce their intraocular pressure. The patients were asked to have a follow-up visit at 1 week as well as 1, 3, 6, and 12 months postoperatively. Evaluations included BCVA, IOP and slit-lamp examination. Central graft thickness (CGT) was measured using anterior segment optical coherence tomography (AS-OCT; RTVue, Optovue Inc., Fremont, CA, USA) at each visit after surgery. The graft failure was defined as loss of clarity and irreversible edema in a graft which was previously recorded to be clear and thin [21].

Postoperatively, 3 images of the central endothelium were captured at each visit using confocal microscope (Heidelberg Retinal Tomograph 3 with Rostock Cornea Module; Heidelberg Engineering GmbH, Heidelberg, Germany). The ECD was determined using the standard frame method [22], assessed from the frame with clear image of endothelium selected from several passes through the endothelium in scans of each cornea (Appendix A). All cells inside the frame were identified by using a point-and-click method, which marked each cell so as to avoid repeated count of cells (Appendix A). Then ECD was determined by the Rostock Cornea Module in each image automatically. To reduce sampling error, all the endothelial cell images were assessed by one masked investigator and the final ECD at each visit was the average of 3 central counts. The representative endothelial images in the two groups are shown in Appendix A. Endothelial cell loss (ECL) percentage was calculated by subtracting the ECD at each follow-up from the donor ECD, then divided by the donor ECD and multiplied by 100.

Statistical analysis was performed with SPSS software version 20.0 (SPSS, Inc., Chicago, IL, USA). Continuous variables were reported as mean ± SD. Frequency distributions and percentages were used for categorical variables. Continuous variables, such as age, preoperative BCVA, donor endothelial cell count, postoperative BCVA, CGT, ECD, and ECL between the injector group and Busin group, were assessed by independent sample t-test. A paired sample t-test was used to analyze whether there was any difference between preoperative and postoperative BCVAs. Categorical variables such as sex ratio, proportion of glaucoma surgery history and so on were analyzed with chi-square test. A *p* value < 0.05 was defined as statistically significant.

## 3. Results

Twenty-four eyes of 24 patients with ICE syndrome were included in this study. Twelve eyes received Busin glide-assisted DSAEK, and the other 12 had injector-assisted DSAEK. Demographic, preoperative, and operative data are presented in Table 1. All patients were followed up for at least 12 months. Eyes in the Busin group had more severe iris damage than the injector group according to the slit-lamp microscope examination, as shown in Appendix A.

*Visual outcomes*: For the total 24 cases, the BCVA improved from the preoperative 0.99 ± 0.61 to 0.36 ± 0.35 at 12 months after operation (*p* < 0.001). There was no significant difference in the preoperative BCVA between the injector group and Busin group (*p* = 0.512), and no significant difference was found as well in the BCVA at 12 months after surgery (*p* = 0.311). However, the BCVA in the injector group significantly improved by 0.63 ± 0.47 (*p* = 0.001), while the improvement in the Busin group was 0.64 ± 0.49 (*p* = 0.013). There was no statistical difference in BCVA variation between the two groups (*p* = 0.933).

*Central graft thickness*: Postoperative AS-OCT central graft thickness measurements were available for 24 patients. The CGT in the injector group was 94.56 ± 20.23 μm at one month, 90.03 ±15.4 μm at three months, 88.62 ± 18.56 μm at six months, and 89.68 ± 16.32 μm at 12 months after surgery. The CGT in the Busin group was 100.4.3 ± 22.39 μm at one month, 95.62 ± 19.83 μm at three months, 96.2 ± 19.32 μm at six months, and 97.07 ± 18.62 μm at 12 months after surgery. The postoperative GCT did not differ between the 2 study groups at each visit (all *p* > 0.5). Figure 4 shows representative AS-OCT images of eyes at 1 and 12 months after DSAEK in the two groups.

*Donor endothelial cell loss*: The postoperative ECD and postoperative ECL data are presented in Table 2. The preoperative donor ECD was 3003 ± 398 cells/mm^2^ in the injector group and 3144 ± 291 cells/mm^2^ in the Busin group (*p* = 0.334). One month postoperatively, the ECD in the injector group was 2341 ± 519 cells/mm^2^ with an ECL of 21.80 ± 15.01%, significantly lower than 33.69 ± 9.75% in the Busin group (*p* = 0.031). In the later follow-ups at 3, 6 and 12 months, the ECD in the injection group kept being greater than that of the Busin group, with lower ECL correspondingly. However, the differences were not statistically different (*p* ≥ 0.115). Figure 5 shows the postoperative survival ratio of endothelial cells in the two groups.

*Complications*: All surgeries were performed successfully with no intraoperative complications. One patient in the injection group suffered from postoperative graft dislocation and underwent rebubbling, while none in the Busin group had this complication, without statistical difference between the two groups (*p* = 1.000). One graft in the injector group developed rejection 1-week after surgery, which was reversed by topical immunosuppressive agent and corticosteroid, and it presented no sign of graft failure at the last follow-up. Two patients in Busin group had high intraocular pressure during follow-up, which could not be controlled by medications, and finally underwent glaucoma surgery. No other complications occurred in the 24 cases during the follow-up period. Two representative cases in the two groups are shown in Figure 6.

## 4. Discussion

Endothelial keratoplasty is the preferred treatment for ICE patients. Although DMEK has lower rejection and better postoperative visual acuity, it is still secondary to DSAEK in China since DMEK has high requirements for donor materials, and a long learning curve for surgeons [23]. Previous studies demonstrated that DSAEK was an effective way to treat ICE syndrome [5,11,13], although it is quite difficult to perform the operation because of the peripheral anterior synechiae formation, iris abnormalities, and shallow anterior chamber [24].

In the procedure of DSAEK, the method of graft insertion is a key issue concerning the endothelial cell survival. Various techniques have been reported in abnormal anterior segments, such as suture pull-through technique [25], donor fixation sutures [26], and injector device [11]. The pull-through technique with Busin glide insertion is a popular procedure for DSAEK in the last decade; however, the high ratio of postoperative ECL is a drawback that needs to be improved. Especially for ICE patients, use of the pull-through technique will more possibly cause iris damage, because the iris hole could intervene the flow of aqueous humor and produce the mechanical stress on the corneal endothelium [27]. It was reported that the graft was compressed when it was pulled through the corneal incision, which might be the major reason that caused postoperative ECL [16,28]. Terry et al. suggested a method for tissue insertion without any damaging compression from the incision [28], in which the graft was drawn into the tube of the injector and directly injected into the AC. It was a less invasive technique with no direct compression from the wound, which can greatly reduce the injury to the corneal endothelial cells. In addition, the insertion of the donor graft by forceps may make the unfolding of the donor graft difficult in AC, which may need further mechanical handling and increase the possibility of central cell loss [28,29,30]. In contrast, the injector permits the graft to enter the AC smoothly without need of pulling of the forceps, and makes it unfold automatically with or without the help of BSS. These two reasons were considered as the principal ones for the difference in endothelial cell survival between the injector and Busin glide groups in our results.

After DSAEK, corneal edema relieved in 1 to 3 months, and BCVA improved significantly both in the injector and Busin group at 12 months, which is similar to the results reported in previous studies [9,24,31]. Thus, these two graft injection techniques can effectively improve the visual acuity for ICE syndrome patients.

Our results showed there was a significant decrease in ECD at 1 month after surgery, suggesting that the most significant ECL in DSAEK grafts occurred at the early postoperative period, which is in accordance with the previous reports [32]. For ICE patients treated with DSAEK, the ECL ratio varied from 37% to 43.7% at 1 month [9,13], which is similar to the ratio of 33.69% in the Busin group in our results. However, our injector group presented a significantly lower ECL ratio of 21.80%, which is better than the previous reports [9,13]. It was acknowledged that the complicated anterior segment in eyes with ICE syndrome usually leads to great difficulty in the process of graft insertion and positioning [9,11,24,31,33], which is an important reason for the increased preoperative ECL ratio [9]. Our results indicate that the injector technique can effectively decrease the intraoperative endothelial cell damage and postoperative ECL. At 3, 6 and 12 months after surgery, the survival rate of endothelial cells in the injection group was greater than that in the Busin group, but there was no statistical difference. In the future, prospective design research and a larger sample size are needed to compare the clinical outcomes between the two different techniques.

Long-term graft survival after DSAEK was poor in the eyes with ICE syndrome. In the previous 3 studies, the ratio of secondary graft failure was 33% (4 of 12 grafts), 55% (11 of 20), and 78% (7 of 9), and the mean time of graft survival was 56 months, 23.4 months, and 19 months, respectively [9,11,13]. In the current study, no one case developed graft failure during the 12-month follow-up period. One case in the injector group experienced rejection 1 week postoperatively which was later reversed by topical immunosuppressive agent and corticosteroid, and it had no sign of graft failure at the last follow-up. Previous study has reported that among ICE patients with graft failure, postoperative rebubbling is more common, suggesting that rebubbling is perhaps associated with postoperative ECL [13]. In our study, there was no statistical difference between the two groups in the incidence of postoperative rebubbling, which indicated that the postoperative ECL difference of the two groups was not related to rebubbling in the current study. The prekeratoplasty glaucoma surgery is an independent risk factor for graft failure after DSAEK [34,35,36]. In our study, the prevalence of prekeratoplasty glaucoma surgery (3/24, 12.5%) is lower than previous studies (34.5~77%) [11], which may be attributable to the high ratio of graft survival in our results. In addition, the shorter term of follow-up in the current study also contributes to the better outcome of graft survival, which urges us to carry on with longer-term investigation.

In conclusion, the current study demonstrated that DSAEK with the injector for graft insertion was a safe and efficient technique for the treatment of eyes with ICE syndrome. The endothelial cell survival was better compared to the pulling-through technique with Busin glide in the early postoperative period. The injector allowed the graft to enter the anterior chamber safely without the use of AC irrigation, and increased the efficiency of graft unfolding and attachment.

## Figures and Tables

**Figure 1 jcm-12-01856-f001:**
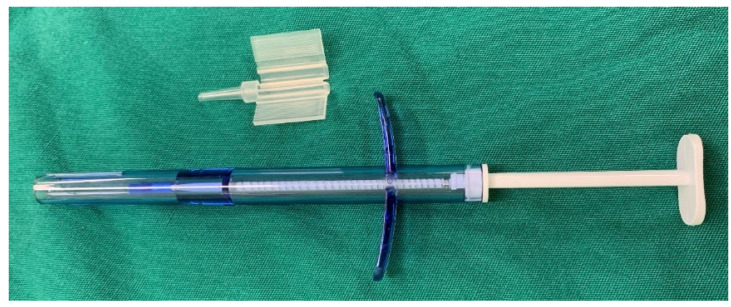
The intraocular lens injector for graft insertion during Descemet stripping automated endothelial keratoplasty.

**Figure 2 jcm-12-01856-f002:**
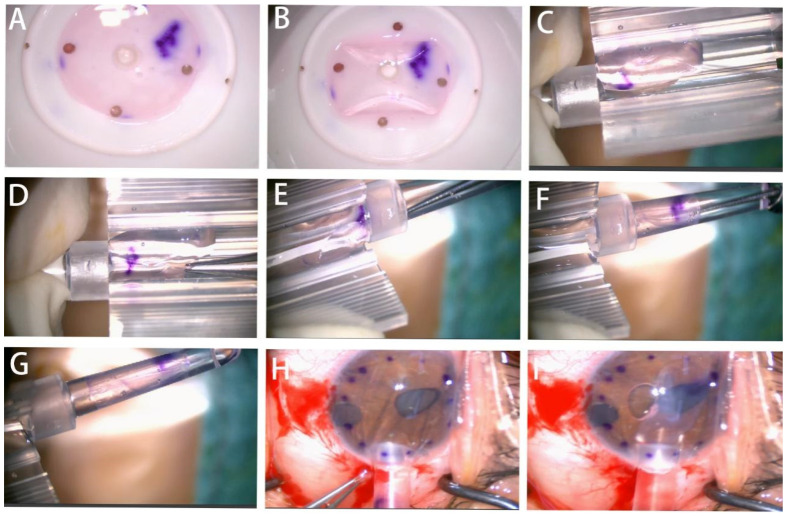
Endothelial graft lenticule delivery using the injector. (**A**,**B**) Make the endodermis of donor facing upward and then fold endothelium. (**C**,**D**) Put the graft into the placement part of the injector, make the corneal endothelium side up and be immersed in the balanced salt solution (BSS), then adjust the position of the graft. (**E**–**G**) The extension part of the endothelial hook is used to extend into the injection tube of the injector, and the graft in the placement part is slowly and completely pulled into the injection tube through the contact end of the extension part. (**H**) The nozzle of the injector is inserted through the temporal incision. (**I**) The graft is delivered into the anterior chamber through the incision.

**Figure 3 jcm-12-01856-f003:**
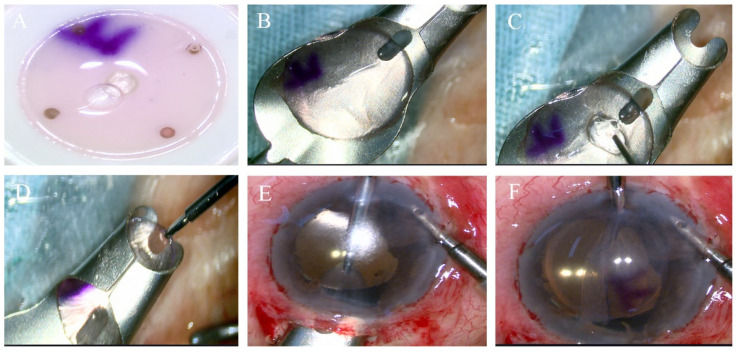
Endothelial graft lenticule delivery using the Busin glide. (**A**) The endodermis of donor is facing upward. (**B**,**C**) Put the graft into the placement part of the glide, make the corneal endothelium side up and a small amount of viscoelastic was placed on the endothelial surface. (**D**–**F**) The glide was inverted and positioned at the entrance of the corneal tunnel, and the Busin forceps were passed to grasp the edge of donor lenticule and to pull it into the AC.

**Figure 4 jcm-12-01856-f004:**
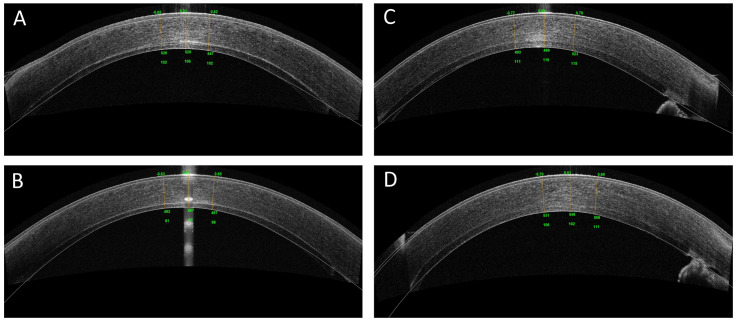
OCT imaging of the cornea after DSAEK. (**A**,**B**) OCT imaging of the cornea in the injector group at 1 month (106 μm) and 12 months (89 μm) respectively after DSAEK. (**C**,**D**) OCT imaging of the cornea in the Busin group at 1 month (110 μm) and 12 months (102 μm) respectively after DSAEK.

**Figure 5 jcm-12-01856-f005:**
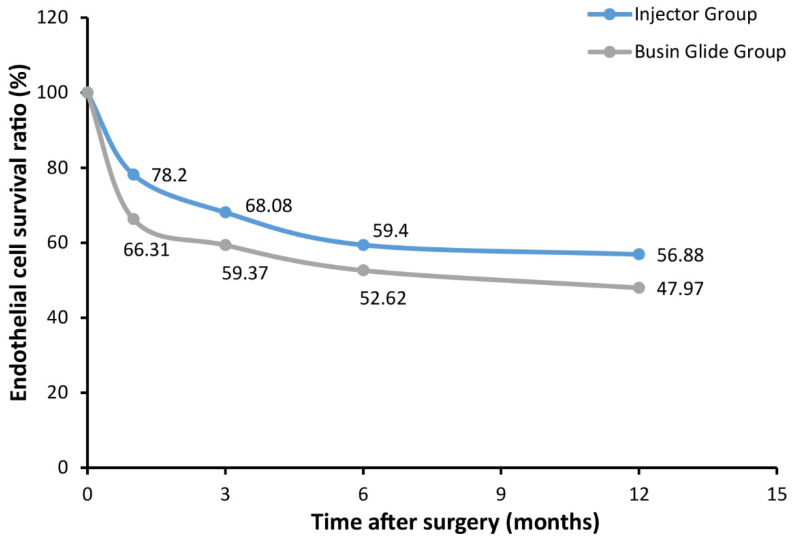
Mean percentage of endothelial cell density in the Injector and Busin groups. 1, 3, 6 and 12 months after DSAEK for iridocorneal endothelial syndrome patients.

**Figure 6 jcm-12-01856-f006:**
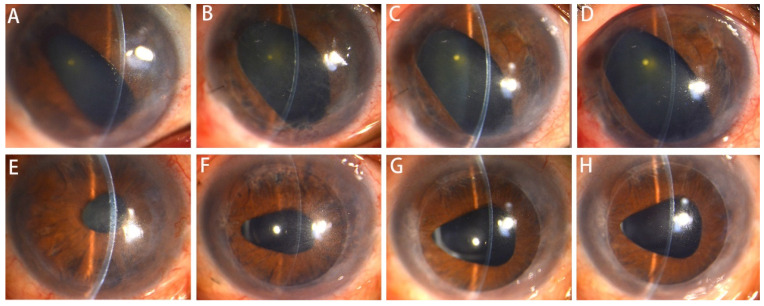
Slit lamp photographs of the two patients who underwent DSAEK with different graft insertion techniques. (**A**–**D**) A representative patient who underwent DSEAK using the injector. Photos A to D respectively show the anterior segment of the patient before operation, 1 month, 6 months and 1 year after operation. (**E**–**H**) A representative patient who underwent DSEAK using the Busin glide. Photos E to H, respectively, show the anterior segment of the patient before operation, 1 month, 6 months and 1 year after operation.

**Table 1 jcm-12-01856-t001:** Clinical and demographic characteristics of patients who received DSAEK using the injector or the Busin glide techniques.

	Injector (*n* = 12)	Busin Glide (*n* = 12)	*p* Value
Age (years)	55.00 ± 10.55	52.84 ± 18.00	0.723
Sex (male/female)	5/7	6/6	0.682
Preoperative BCVA (LogMAR)	0.91 ± 0.68	1.08 ± 0.53	0.512
History of glaucoma surgery	2 (16.67%)	1 (8.33%)	0.537
History of keratoplasty	0	1 (8.33%)	0.307
Donor endothelial cell count (cells/mm^2^)	3003 ± 398	3144 ± 291	0.334
Type of surgery			
DSAEK alone	4 (33.33%)	4 (33.33%)	1.000
DSAEK + cataract surgery + IOL	8 (66.67%)	8 (66.67%)	1.000

DSAEK = Descemet stripping automated endothelial keratoplasty; BCVA = best-corrected visual acuity; LogMAR = logarithm of minimal angle of resolution; IOL = intraocular lens.

**Table 2 jcm-12-01856-t002:** Donor and postoperative ECD and postoperative ECL in eyes that underwent DSAEK using the injector or the Busin glide techniques.

	Injector (*n* = 12)	Busin Glide (*n* = 12)	*p* Value
Donor ECD (cells/mm^2^)	3003 ± 398	3144 ± 291	0.334
ECD at 1 month (cells/mm^2^)	2341 ± 519	2095 ± 393	0.203
ECL at 1 month (%)	21.80 ± 15.01	33.69 ± 9.75	0.031
ECD at 3 months (cells/mm^2^)	2039 ± 487	1874 ± 374	0.364
ECL at 3 months (%)	31.92 ± 15.29	40.63 ± 10.23	0.115
ECD at 6 months (cells/mm^2^)	1787 ± 501	1669 ± 481	0.561
ECL at 6 months (%)	40.60 ± 14.29	47.38 ± 13.60	0.247
ECD at 12 months (cells/mm^2^)	1713 ± 557	1521 ± 471	0.370
ECL at 12 months (%)	43.12 ± 16.03	52.03 ± 13.34	0.153

DSAEK = descemet stripping automated endothelial keratoplasty; ECD = endothelial cell density; ECL = endothelial cell loss.

## Data Availability

Data sharing is not applicable to this article.

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
