# Peer review of "Clinical Outcomes of the Intraocular Lens Injector and Busin Glide for Descemet Stripping Automated Endothelial Keratoplasty in Patients with Iridocorneal Endothelial Syndrome"

_jcm, 2023, doi:10.3390/jcm12051856_

Round 1

Reviewer 1 Report

I commend the authors for addressing this important topic. Insertion techniques for endothelial grafts remain of paramount importance for graft survival, especially when preloaded grafts are not readily available. The authors conducted their study on patients with ICE, a rare entity that complicates surgical outcomes. I have a few comments that I would like the authors to address:

1- The English language needs to be revised for a few typographic errors.

2- I do not believe the authors described their study design correctly. This is not a case-control study, it is an interventional comparative study. The fact that the study compares 2 different surgical techniques should be reflected in the title of the manuscript as well.

3- Patient allocation into 2 equal groups should be stated clearly at the beginning of the Patients and Methods before detailing the surgical techniques.

4- The orientation of the injector (bevel-up vs bevel-down) during graft insertion should be clarified.

5- I suggest adding a composite figure detailing the surgical technique utilizing the Busin glide.

6- There was practically no difference between both techniques regarding ECL, except at the 1-month visit, a difference that is not clinically relevant. The Conclusions should be rewritten to reflect these results.

Author Response

1- The English language needs to be revised for a few typographic errors.

Response:Thank you very much for your comments. We have corrected the typographic errors in the revised manuscript.

2- I do not believe the authors described their study design correctly. This is not a case-control study, it is an interventional comparative study. The fact that the study compares 2 different surgical techniques should be reflected in the title of the manuscript as well.

Response:Thank you very much for discovering the errors. We have changed the description of case-control study into interventional comparative study in the revised manuscript. In addition, the title has been corrected to reflect that we compared 2 different surgical techniques.

3- Patient allocation into 2 equal groups should be stated clearly at the beginning of the Patients and Methods before detailing the surgical techniques.

Response:We have added the information of patient grouping in the methods of revised manuscript: According to the method of graft insertion in DSAEK (intraocular lens injector or Busin glide), we divided the patients included into the injector group and the Busin group.

4- The orientation of the injector (bevel-up vs bevel-down) during graft insertion should be clarified.

Response:We have added the orientation of the injector in the methods of revised manuscript: The injector inclined downward into the AC with graft endothelial side up. Then the injector rotated 90 degrees clockwise and slowly pushed the folded graft into the AC.

5- I suggest adding a composite figure detailing the surgical technique utilizing the Busin glide.

Response: We have added the Figure 3 to show the graft insertion technique using the Busin glide in the methods of revised manuscript.

6- There was practically no difference between both techniques regarding ECL, except at the 1-month visit, a difference that is not clinically relevant. The Conclusions should be rewritten to reflect these results.

Response: We have reflected in the revised manuscript that there is a statistical difference in ECL between the injection group and the Busin group at 1 month after operation. At other follow-up time points, although the survival rates of endothelial cell in the injection group were greater than that in the Busin group, no statistical difference was found.

Reviewer 2 Report

Dear authors,

The aim of the study „Clinical outcomes of the intraocular lens injector for DSAEK in patients with iridocorneal endothelial syndrom“ is to report the outcomes of DSAEK performed in ICE syndrome patients using the IOL injector vs. The Busin glide.

This is an interesting topic, with an interesting and rare underlying disease. The authors conclude, that endothelial cell loss 1 month after DSAEK was significantly lower in the group with the injector compared to the Busin glide. However, these differences were no longer statistically significant in the further course. Please point out in the discussion that although a difference could be demonstrated, it was no longer statistically significant.

The authors compared the two groups and found only a statistically significant difference in endothelial cell loss. Rebubbling may also cause ECL. Please provide information about the number of rebubblings performed in the two groups, compare them statistically and add this to the discussion. 

Thank you

Author Response

      Thank you for your comments. We have reflected in the discussion of revised manuscript that there is a statistical difference in ECL between the injection group and the Busin group at 1 month after operation. At other follow-up time points, although the survival rates of endothelial cell in the injection group were greater than that in the Busin group, no statistical difference was found.

     We counted the number of patients with rebubbling between the two groups. There was 1 case in the injection group and no case in the Busin group. There was no statistical difference between the two groups (P=1.000), and we have added it to the results and discussion of the revised manuscript.